# Efficient transgenesis and annotated genome sequence of the regenerative flatworm model *Macrostomum lignano*

Jakub Wudarski[1], Daniil Simanov[1,2], Kirill Ustyantsev[3], Katrien de Mulder[2,8], Margriet Grelling[1], Magda Grudniewska [1], Frank Beltman[1], Lisa Glazenburg[1], Turan Demircan[2,9], Julia Wunderer[4], Weihong Qi[5], Dita B. Vizoso[6], Philipp M. Weissert[1], Daniel Olivieri[1,10], Stijn Mouton[1], Victor Guryev[1], Aziz Aboobaker [7], Lukas Schärer[6], Peter Ladurner[4] & Eugene Berezikov [1,2,3]

Regeneration-capable flatworms are informative research models to study the mechanisms of stem cell regulation, regeneration, and tissue patterning. However, the lack of transgenesis methods considerably hampers their wider use. Here we report development of a transgenesis method for *Macrostomum lignano*, a basal flatworm with excellent regeneration capacity. We demonstrate that microinjection of DNA constructs into fertilized one-cell stage eggs, followed by a low dose of irradiation, frequently results in random integration of the transgene in the genome and its stable transmission through the germline. To facilitate selection of promoter regions for transgenic reporters, we assembled and annotated the *M. lignano* genome, including genome-wide mapping of transcription start regions, and show its utility by generating multiple stable transgenic lines expressing fluorescent proteins under several tissue-specific promoters. The reported transgenesis method and annotated genome sequence will permit sophisticated genetic studies on stem cells and regeneration using *M. lignano* as a model organism.

[1] European Research Institute for the Biology of Ageing, University of Groningen, University Medical Center Groningen, Antonius Deusinglaan 1, 9713AV Groningen, The Netherlands. [2] Hubrecht Institute-KNAW and University Medical Centre Utrecht, Uppsalaan 8, 3584CT Utrecht, The Netherlands. [3] Institute of Cytology and Genetics, Prospekt Lavrentyeva 10, 630090 Novosibirsk, Russia. [4] Institute of Zoology and Center for Molecular Biosciences Innsbruck, University of Innsbruck, Technikerstr. 25, A-6020 Innsbruck, Austria. [5] Functional Genomics Center Zurich, Winterthurerstrasse 190, Zurich CH-8057, Switzerland. [6] Evolutionary Biology, Zoological Institute, University of Basel, Vesalgasse 1, CH-4051 Basel, Switzerland. [7] Department of Zoology, University of Oxford, Tinbergen Building, South Parks Road, Oxford OX1 3PS, United Kingdom. [8] Present address: Molecular laboratory, AZ St. Lucas Hospital, Gent 9000, Belgium. [9] Present address: Department of Medical Biology, International School of Medicine, İstanbul Medipol University, Istanbul 34810, Turkey. [10] Present address: Friedrich Miescher Institute for Biomedical Research, Maulbeerstrasse 66, Basel CH-4058, Switzerland. Correspondence and requests for materials should be addressed to E.B. (email: e.berezikov@umcg.nl)

Animals that can regenerate missing body parts hold clues to advancing regenerative medicine and are attracting increased attention[1]. Significant biological insights on stem cell biology and body patterning were obtained using free-living regeneration-capable flatworms (Platyhelminthes) as models[2–4]. The most often studied representatives are the planarian species *Schmidtea mediterranea*[2] and *Dugesia japonica*[5]. Many important molecular biology techniques and resources are established in planarians, including fluorescence-activated cell sorting, gene knockdown by RNA interference, in situ hybridization, and genome and transcriptome assemblies[4]. One essential technique still lacking in planarians; however, is transgenesis, which is required for in-depth studies involving e.g., gene over-expression, dissection of gene regulatory elements, real-time imaging and lineage tracing. The reproductive properties of planarians, including asexual reproduction by fission and hard non-transparent cocoons containing multiple eggs in sexual strains, make development of transgenesis technically challenging in these animals.

More recently, a basal flatworm *Macrostomum lignano* (Macrostomorpha) emerged as a model organism that is complementary to planarians[6–9]. The reproduction of *M. lignano*, a free-living marine flatworm, differs from planarians, as it reproduces by laying individual fertilized one-cell stage eggs. One animal lays ~1 egg per day when kept in standard laboratory conditions at 20 °C. The eggs are around 100 microns in diameter, and follow the archoophoran mode of development, having yolk-rich oocytes instead of supplying the yolk to a small oocyte via yolk cells[10]. The laid eggs have relatively hard shells and can easily be separated from each other with the use of a fine plastic picker. These features make *M. lignano* eggs easily amenable to various manipulations, including microinjection[11]. In addition, *M. lignano* has several convenient characteristics, such as ease of culture, transparency, small size, and a short generation time of three weeks[6,7]. It can regenerate all tissues posterior to the pharynx, and the rostrum[12]. This regeneration ability is driven by stem cells, which in flatworms are called neoblasts[3,4,13]. Recent research in planarians has shown that the neoblast population is heterogeneous and consists of progenitors and stem cells[14,15]. The true pluripotent stem cell population is, however, not identified yet.

Here we present a method for transgenesis in *M. lignano* using microinjection of DNA into single-cell stage embryos and demonstrate its robustness by generating multiple transgenic tissue-specific reporter lines. We also present a significantly improved genome assembly of the *M. lignano* DV1 line and an accompanying transcriptome assembly and genome annotation. The developed transgenesis method, combined with the generated genomic resources, will enable new research avenues on stem cells and regeneration using *M. lignano* as a model organism, including in-depth studies of gene overexpression, dissection of gene regulatory elements, real-time imaging and lineage tracing.

## Results

**Microinjection and random integration of transgenes.** *M. lignano* is an obligatorily non-self-fertilizing simultaneous hermaphrodite (Fig. 1a) that produces substantial amounts of eggs (Fig. 1b, c). We reasoned that microinjection approaches used in other model organisms, such as *Drosophila*, zebrafish and mouse, should also work in *M. lignano* eggs (Fig. 1d, Supplementary Movie 1). First, we tested how the egg handling and micro-injection procedure itself impacts survival of the embryos (Supplementary Table 1). Separating the eggs laid in clumps and transferring them into new dishes resulted in a 17% drop in hatching rate, and microinjection of water decreased survival by a further 10%. Thus, in our hands >70% of the eggs can survive the microinjection procedure (Supplementary Table 1). When we injected fluorescent Alexa 555 dye, which can be used to track the injected material, about 50% of the eggs survived (Supplementary Table 1). For this reason, we avoided tracking dyes in subsequent experiments. Next, we injected in vitro synthesized mRNA encoding green fluorescent protein (GFP) and observed its expression in all successfully injected embryos (*n* > 100) within 3 h after injection (Fig. 1e), with little to no autofluorescence detected in either embryos or adult animals (Supplementary Fig. 1). The microinjection technique can thus be used to deliver biologically relevant materials into single-cell stage eggs with a manageable impact on the survival of the embryos.

To investigate whether exogenous DNA constructs can be introduced and expressed in *M. lignano*, we cloned a 1.3 kb promoter region of the translation elongation factor 1 alpha (*EFA*) gene and made a transcriptional GFP fusion in the Minos transposon system (Supplementary Fig. 2a). Microinjection of the *Minos::pEFA::eGFP* plasmid with or without Minos transposase mRNA resulted in detectable expression of GFP in 5–10% of the injected embryos (Supplementary Fig. 2c). However, in most cases GFP expression was gradually lost as the animals grew (Supplementary Fig. 2f), and only a few individuals transmitted the transgene to the next generation. From these experiments we

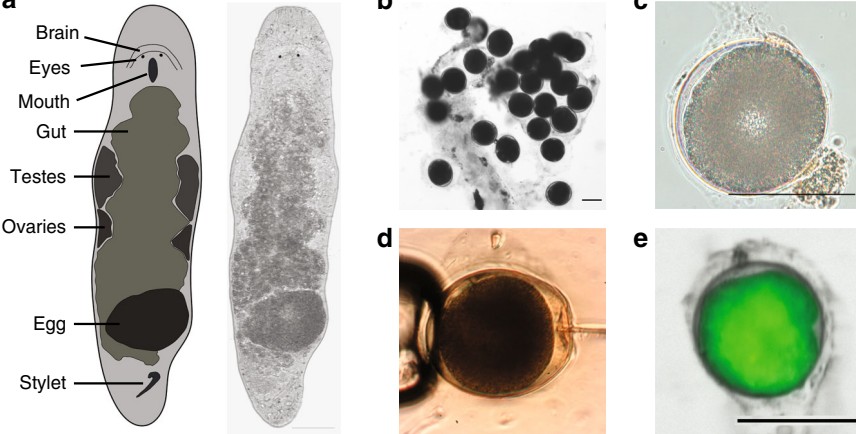

**Fig. 1** *Macrostomum lignano* embryos are amenable to microinjection. **a** Schematic morphology and a bright-field image of an adult *M. lignano* animal. **b** Clump of fertilized eggs. **c** DIC image of a one-cell stage embryo. **d** Microinjection into a one-cell stage embryo. **e** Expression of GFP in the early embryo 3 h after injection with in vitro synthesized *GFP* mRNA. Scale bars are 100 μm

**Table 1 Efficiency of transgenesis with different reporter constructs and treatments**

| Reporter | Injected line | Injected DNA | Irradiation treatment | Injected eggs | Positive hatchlings (%) | Germline transmission (%) | Established lines |
|---|---|---|---|---|---|---|---|
| EFA::eGFP | DV1 | PCR | — | 269 | 39 (14.50) | 5 (1.86) | NL1 |
| EFA::oGFP | DV1 | Plasmid | — | 114 | 28 (24.56) | 0 | — |
| EFA::oGFP | DV1 | Plasmid | 2.5 Gy | 42 | 13 (30.95) | 2 (4.76) | — |
| EFA::oGFP | DV1 | Fragment | 2.5 Gy | 102 | 4 (3.92) | 2 (1.96) | NL7 |
| EFA::oCherry | DV1 | Plasmid | 2.5 Gy | 80 | 4 (5.00) | 1 (1.25) | NL3 |
| EFA::oCherry | DV1 | Fragment | 2.5 Gy | 36 | 6 (16.67) | 3 (8.33) | NL4, NL5, NL6 |
| EFA::H2B::oGFP | DV1 | Fragment | 2.5 Gy | 38 | 10 (26.32) | 2 (5.26) | NL20 |
| ELAV4::oGFP | DV1 | Fragment | 2.5 Gy | 56 | 29 (51.79) | 2 (3.57) | NL21 |
| MYH6::oGFP | DV1 | Fragment | 2.5 Gy | 103 | 13 (12.62) | 1 (0.97) | NL9 |
| APOB::oGFP | DV1 | Fragment | 2.5 Gy | 65 | 2 (3.08) | 1 (1.54) | NL22 |
| CABP7::oGFP | DV1 | Plasmid | — | 20 | 2 (10.00) | 1 (5.00) | NL23 |
| CABP7::oNeon Green; ELAV4::oScarlet-I | NL10 | Plasmid | — | 137 | 3 (2.19) | 2 (1.46) | NL24 |

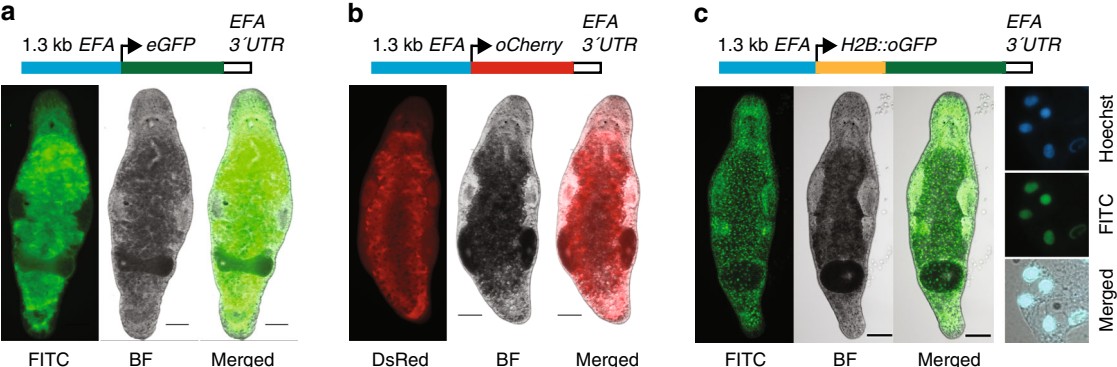

**Fig. 2** Ubiquitously expressed elongation factor 1 alpha promoter transgenic lines. **a** NL1 line expressing enchanced GFP (eGFP). **b** NL3 line expressing codon-optimized Cherry (oCherry). **c** NL20 line expressing codon-optimized nuclear localized H2B::oGFP fusion. Right column—single cells from a macerated animal showing nuclear localization of GFP. FITC—FITC channel; DsRed—DsRed channel; BF—bright-field; Hoechst—DNA staining by Hoechst. Scale bars are 100 μm

established the HUB1 transgenic line with ubiquitous GFP expression, which recapitulates expression of the *EFA* gene determined by in situ hybridization (Supplementary Fig. 2d, e). Stable transgene transmission in the HUB1 line has been observed for over 50 generations[16,17].

The expected result for transposon-mediated transgenesis is genomic integration of the fragment flanked by transposon inverted terminal repeats. However, plasmid sequences outside the terminal repeats, including the ampicillin resistance gene, were detected in the HUB1 line, suggesting that the integration was not mediated by Minos transposase. Furthermore, southern blot analysis revealed that HUB1 contains multiple transgene copies (Supplementary Fig. 2g). We next tried a different transgenesis strategy using meganuclease *I-SceI*[18] to improve transgenesis efficiency (Supplementary Fig. 2b). We observed a similar 3–10% frequency of initial transgene expression, and only two instances of germline transmission, one of which resulted from the negative control experiment without co-injected meganuclease protein (Supplementary Fig. 2c). These results suggest that *I-SceI* meganuclease does not increase efficiency of transgenesis in *M. lignano*, but instead that exogenous DNA can be integrated in the genome by non-homologous recombination using the endogenous DNA repair machinery.

**Improvement of integration efficiency.** The frequency of germline transgene transmission in the initial experiments was

<0.5% of the injected eggs, while transient transgene expression was observed in up to 10% of the cases (Supplementary Fig. 2c, f). We hypothesized that mosaic integration or mechanisms similar to extrachromosomal array formation in *C. elegans*[19] might be at play in cases of transient gene expression in *M. lignano*. We next tested two approaches used in *C. elegans* to increase the efficiency of transgenesis: removal of vector backbone and injection of linear DNA fragments[20], and transgene integration by irradiation[19]. Injection of PCR-amplified vector-free transgenes resulted in the germline transmission in 5 cases out of 269 injected eggs, or 1.86% (Table 1), and the stable transgenic line NL1 was obtained during these experiments (Fig. 2a). In this line, the *GFP* coding sequence was optimized for *M. lignano* codon usage. While we did not observe obvious differences in expression levels between codon-optimized and non-optimized *GFP* sequences, we decided to use codon-optimized versions in all subsequent experiments.

*M. lignano* is remarkably resistant to ionizing radiation, and a dose as high as 210 Gy is required to eliminate all stem cells in an adult animal[8,21]. We reasoned that irradiation of embryos immediately after transgene injection might stimulate non-homologous recombination and increase integration rates. Irradiation dose titration revealed that *M. lignano* embryos are less resistant to radiation than adults and that a 10 Gy dose results in hatching of only 10% of the eggs, whereas >90% of eggs survive a still substantial dose of 2.5 Gy (Supplementary Table 2). Irradiating injected embryos with 2.5 Gy resulted in 1–8%

**Table 2 Characteristics of Mlig_3_7 genome assembly**

|  | Contigs | Scaffolds |
|---|---|---|
| Total number | 5980 | 5270 |
| Total length | 762,843,491 | 764,424,962 |
| Average length | 127,565 | 145,052 |
| Shortest | 1370 | 3068 |
| Longest | 2,680,987 | 2,680,987 |
| N50 | 215,172 | 245,921 |

germline transmission rate for various *EFA* promoter constructs in both plasmid and vector-free forms (Table 1). The stable transgenic line NL3 expressing codon-optimized red fluorescent protein Cherry was obtained in this way (Fig. 2b), demonstrating that ubiquitous expression of fluorescent proteins other than GFP is also possible in *M. lignano*. Finally, to test nuclear localization of the reporter protein, we fused GFP with a partial coding sequence of the histone 2B (H2B) gene as described previously[22]. The injection of the transgene fragment followed by irradiation demonstrated 5% transgenesis efficiency (Table 1), and the stable NL20 transgenic line with nuclear GFP localization was established (Fig. 2c).

**Genome assembly and annotation**. To extend the developed transgenesis approach to promoters of other genes, an annotated genome assembly of *M. lignano* was required. Toward this, we have generated and sequenced 29 paired-end and mate-pair genomic libraries of the DV1 line using 454 and Illumina technologies (Supplementary Table 3). Assembling these data using the MaSuRCA genome assembler[23] resulted in a 795 Mb assembly with N50 scaffold size of 11.9 kb. While this assembly was useful for selecting several novel promoter regions, it suffered from fragmentation. In a parallel effort, a PacBio-based assembly of the DV1 line, termed ML2, was recently published[9]. The ML2 assembly is 1040 Mb large and has N50 contig size of 36.7 kb and NG50 contig size of 64.5 kb when adjusted to the 700 Mb genome size estimated from k-mer frequencies[9]. We performed fluorescence-based genome size measurements and estimated that the haploid genome size of the DV1 line is 742 Mb (Supplementary Fig. 3d,e,f). It was recently demonstrated that *M. lignano* can have a polymorphic karyotype, where in addition to the basal $2n = 8$ karyotype, also animals with aneuploidy for the large chromosome, with $2n = 9$ and $2n = 10$ exist[24]. We confirmed that our laboratory culture of the DV1 line has predominantly $2n = 10$ and $2n = 9$ karyotypes (Supplementary Fig. 3a, b) and estimated that the size of the large chromosome is 240 Mb (Supplementary Fig. 3f). In contrast, an independently established *M. lignano* wild-type line NL10 has the basal karyotype $2n = 8$ and does not show detectable variation in chromosome number (Supplementary Fig. 3c,d). This line, however, was established only recently and was not a part of the genome sequencing effort.

We re-assembled the DV1 genome from the generated Illumina and 454 data and the published PacBio data[9] using the Canu assembler[25] and SSPACE scaffolder[26]. The resulting Mlig_3_7 assembly is 764 Mb large with N50 contig and scaffold sizes of 215.2 Kb and 245.9 Kb, respectively (Table 2), which is greater than threefold continuity improvement over the ML2 assembly. To compare the quality of the ML2 and Mlig_3_7 assemblies, we used the genome assembly evaluation tool REAPR, which identifies assembly errors without the need for a reference genome[27]. According to the REAPR analysis, the Mlig_3_7 assembly has 63.95% of error-free bases compared to 31.92% for the ML2 assembly and 872 fragment coverage distribution (FCD) errors within contigs compared to 1871 in the ML2 assembly (Supplementary Fig. 4a). Another genome assembly evaluation

tool, FRCbam, which calculates feature response curves for several assembly parameters[28], also shows better overall quality of the Mlig_3_7 assembly (Supplementary Fig. 4b). Finally, 96.9% of transcripts from the de novo transcriptome assembly MLRNA150904[8] can be mapped on Mlig_3_7 (>80% identity, >95% transcript length coverage), compared to 94.88% of transcripts mapped on the ML2 genome assembly, and among the mapped transcripts more have intact open reading frames in the Mlig_3_7 assembly than in ML2 (Supplementary Fig. 4c). Based on these comparisons, the Mlig_3_7 genome assembly represents a substantial improvement in both continuity and base accuracy over the ML2 assembly.

More than half of the genome is repetitive, with LTR retrotransposons and simple and tandem repeats accounting for 21 and 15% of the genome, respectively (Supplementary Table 4). As expected from the karyotype of the DV1 line, which has additional large chromosomes, the Mlig_3_7 assembly has substantial redundancy, with 180 Mb in duplicated non-repetitive blocks that are longer than 500 bp and at least 95% identical. When repeat-annotated regions are included in the analysis, the duplicated fraction of the genome rises to 312 Mb.

Since genome-guided transcriptome assemblies are generally more accurate than de novo transcriptome assemblies, we generated a new transcriptome assembly based on the Mlig_3_7 genome assembly using a combination of the StringTie[29] and TACO[30] transcriptome assemblers, a newly developed TBONE gene boundary annotation pipeline, previously published RNA-seq datasets[8,31] and the de novo transcriptome assembly MLRNA150904[8]. Since many *M. lignano* transcripts are trans-spliced[8,9], we extracted reads containing trans-splicer leader sequences from raw RNA-seq data and mapped them to the Mlig_3_7 genome assembly after trimming the trans-splicing parts. This revealed that many more transcripts in *M. lignano* are trans-spliced than was previously appreciated from de novo transcriptome assemblies (6167 transcripts in Grudniewska et al.[8], 7500 transcripts in Wasik et al.[9], 28,273 in this study, Table 3). We also found that almost 7% of the assembled transcripts are in fact precursor mRNAs, i.e., they have several trans-splicing sites and encode two or more proteins (Table 3, Supplementary Fig. 5a). Therefore, in the transcriptome assembly we distinguish between transcriptional units and genes transcribed within these transcriptional units. For this, we developed computational pipeline TBONE (Transcript Boundaries based ON experimental Evidence), which relies on experimental data, such as trans-splicing and polyadenylation signals derived from RNA-seq data, to 'cut' transcriptional units and establish boundaries of mature mRNAs (Supplementary Fig. 5a). The new genome-guided transcriptome assembly, Mlig_RNA_3_7_DV1.v1, has 66,777 transcriptional units, including duplicated copies and alternative forms, which can be collapsed to 33,715 non-redundant transcripts when clustered by 95% global sequence identity (Table 3). These transcriptional units transcribe 72,846 genes, of which 44,328 are non-redundant, 38.8% are trans-spliced and 79.98% have an experimentally defined poly(A) site (Table 3). The non-redundant transcriptome has TransRate scores of 0.4360 and 0.4797 for transcriptional units and gene sequences, respectively, positioning it among the highest quality transcriptome assemblies[32]. The transcriptome is 98.1% complete according to the Benchmarking Universal Single-Copy Orthologs[33], with only 3 missing and 3 fragmented genes (Table 3).

The Mlig_RNA_3_7_DV1 transcriptome assembly, which incorporates experimental evidence for gene boundaries, greatly facilitates selection of promoter regions for transgenesis. Furthermore, we previously generated 5′-enriched RNA-seq libraries from mixed stage populations of animals[8] using RAMPAGE[34]. In our hands, the RAMPAGE signal is not

**Table 3 Characteristics of Mlig_RNA_3_7_DV.v1 transcriptome assembly**

|  | Transcriptional units | Genes |
|---|---|---|
| Number of transcripts | 66,777 | 72,846 |
| Total length | 206 Mb | 182 Mb |
| Number of non-redundant sequences[a] | 33,715 | 44,328 |
| Total length of non-redundant sequences[b] | 127 Mb | 133 Mb |
| Average transcript length | 3.8 kb | 3.0 kb |
| Shortest transcript | 104 nt | 151 nt |
| Longest transcript | 51,585 nt | 47,797 nt |
| Transcripts with single trans-splicing site | 18,894 (28.29%) | 28,273 (38.81%) |
| Transcripts with multiple trans-splicing sites | 4,596 (6.88%) | — |
| Transcripts with defined poly(A) site | 52,707 (78.93%) | 58,259 (79.98%) |
| TransRate score | 0.4360 | 0.4797 |
| Average gene length | 9.4 kb | 7.5 kb |
| Average number of introns per gene | 5.0 | 4.9 |
| Average intron length | 1.4 kb | 1.1 kb |
| Human homolog genes | — | 8006 |
| PFAM domains | — | 5819 |
| Eukaryotic BUSCOs ($n = 303$) |  |  |
| Complete | — | 98.1% |
| Fragmented | — | 1.0% |
| Missing | — | 0.9% |

[a]Sequences with ≥ 95% identity at nucleotide level
[b]Sequences with 100% amino acid identity of ORFs

sufficiently localized around transcription start sites to be used directly by the TBONE pipeline, but it can be very useful for determining transcription starts during manual selection of promoter regions for transgenesis (Supplementary Fig. 5b, c). We used the UCSC genome browser software[35] to visualize genome structure and facilitate design of new constructs for transgenesis (Supplementary Fig. 5). The *M. lignano* genome browser, which integrates genome assembly, annotation and RNA-seq data, is publicly accessible at http://gb.macgenome.org.

**Tissue-specific transgenic lines**. Equipped with the annotated *M. lignano* genome and the developed transgenesis approach, we next set to establish transgenic lines expressing tissue-specific reporters. For this, we selected homologs of the *MYH6*, *APOB*, *ELAV4*, and *CABP7* genes, for which tissue specificity in other model organisms is known and upstream promoter regions can be recognized based on genome annotation and gene boundaries (Supplementary Fig. 5). Similar to the *EFA* promoter, in all cases the transgenesis efficiency was in the range of 1–5% of the injected eggs (Table 1) and stable transgenic lines were obtained (Fig. 3). Expression patterns were as expected from prior knowledge and corroborated by the whole mount in situ hybridization results: the *MYH6::GFP* is expressed in muscle cells, including muscles within the stylet (Fig. 3a, Supplementary Movie 2); *APOB::GFP* is gut-specific (Fig. 3b); *ELAV4::GFP* is testis-specific, including the sperm, which is accumulated in the seminal vesicle (Fig. 3c); and *CABP7::GFP* is ovary-specific and is also expressed in developing eggs (Fig. 3d). Finally, we made a double-reporter construct containing *ELAV4::oNeonGreen* and *CABP7::oScarlet-I* in a single plasmid (Fig. 3e). mNeonGreen[36] and mScarlet[37] are monomeric yellow–green and red fluorescent proteins, respectively, with the highest reported brightness among existing fluorescent proteins. The transgenesis efficiency with the

double-reporter construct was comparable to other experiments (Table 1), and transgenic line NL24 expressing codon-optimized mNeonGreen (oNeonGreen) in testes and codon-optimized mScarlet-I (oScarlet) in ovaries was established (Fig. 3e), demonstrating the feasibility of multi-color reporters in *M. lignano*. The successful generation of stable transgenic reporter lines for multiple tissue-specific promoters validates the robustness of the developed transgenesis method and demonstrates the value of the generated genomic resource.

**Identification of transgene integration sites**. To directly demonstrate that transgenes integrate into the *M. lignano* genome and to establish genomic locations of the integration sites, we initially attempted to identify genomic junctions by inverse PCR with outward-oriented transgene-specific primers (Supplementary Fig. 6a) in the NL7 and NL21 transgenic lines. However, we found that in both cases short products of ~200 nt are preferentially and specifically amplified from genomic DNA of the transgenic lines (Supplementary Fig. 6b, c). The size of the PCR products can be explained by formation of tandem transgenes (Supplementary Fig. 6a), and sequencing confirmed that this is indeed the case (Supplementary Fig. 6d). Next, we used the Genome Walker approach, in which genomic DNA is digested with a set of restriction enzymes, specific adapters are ligated and regions of interest are amplified with transgene-specific and adapter-specific primers. Similarly, many of the resulting PCR products turned out to be transgene tandems. But in the case of the NL21 line we managed to establish the integration site on one side of the transgene (Supplementary Fig. 6e), namely at position 45,440 in scaf3369 (Mlig_3_7 assembly) in the body of a 2-kb long LTR retrotransposon, 10.5 kb downstream from the end of the Mlig003479.g3 gene and 2.5 kb upstream from the start of the Mlig028829.g3 gene.

**Transgene expression in regenerating animals**. Our main rationale for developing *M. lignano* as a new model organism is based on its experimental potential to study the biology of regenerative processes in vivo in a genetically tractable organism. Therefore, it is essential to know whether regeneration could affect transgene stability and behavior. Toward this, we monitored transgene expression during regeneration in the testis- and ovary-specific transgenic lines NL21 and NL23, respectively (Fig. 4). Adult animals were amputated anterior of the gonads and monitored for 10 days. In both transgenic lines regeneration proceeded normally and no GFP expression was observed in the first days of regeneration (Fig. 4). Expression in ovaries was first detected at day 8 after amputation, and in testes at day 10 after amputation (Fig. 4). Thus, tissue-specific transgene expression is restored during regeneration, as expected for a regular genomic locus.

**Discussion**
Free-living regeneration-capable flatworms are powerful model organisms to study mechanisms of regeneration and stem cell regulation[2,4]. Currently, the most popular flatworms among researchers are the planarian species *S. mediterranea* and *D. japonica*[4]. A method for generating transgenic animals in the planarian *Girardia tigrina* was reported in 2003[38], but despite substantial ongoing efforts by the planarian research community it has thus far not been reproduced in either *S. mediterranea* or *D. japonica*. The lack of transgenesis represents a significant experimental limitation of the planarian model systems. Primarily for this reason we focused on developing an alternative, non-planarian flatworm model, *Macrostomum lignano*. We reasoned that the fertilized one-cell stage eggs, which are readily available in this species, will facilitate development of the transgenesis

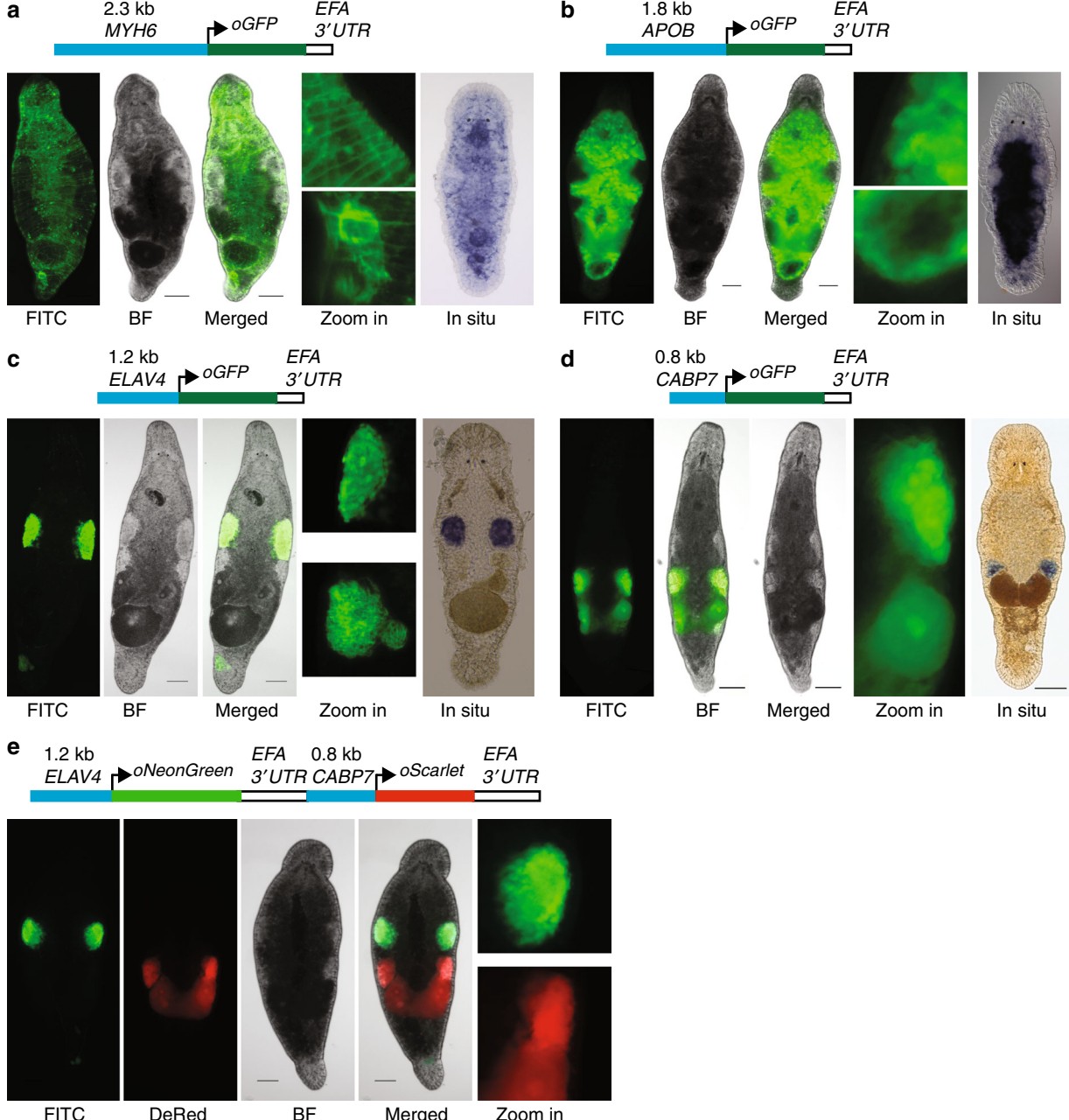

**Fig. 3** Tissue-specific promoter transgenic lines. **a** NL9 line expressing GFP under the muscle-specific promoter of the *MYH6* gene. Zoom in—detailed images of the body wall (top) and stylet (bottom); In situ—whole-mount in situ hybridization expression pattern of *MYH6* transcript. **b** NL22 line expressing GFP under the gut-specific promoter of the *APOB* gene. Zoom in—detailed images of the gut side (top) and distal tip (bottom); In situ—whole-mount in situ hybridization expression pattern of the *APOB* transcript. **c** NL21 line expressing GFP under the testis-specific promoter of the *ELAV4* gene. Zoom in—detailed images of the testis (top) and seminal vesicle (bottom); In situ—whole-mount in situ hybridization expression pattern of the *ELAV4* transcript. **d** NL23 line expressing GFP under the ovary-specific promoter of the *CABP7* gene. Zoom in—detailed image of the ovary and developing egg; In situ—whole-mount in situ hybridization expression pattern of the *CABP7* transcript. **e** NL24 line expressing in a single construct NeonGreen under the testis-specific promoter of the *ELAV4* gene and Scarlet-I under the ovary-specific promoter of the *CABP7* gene. Zoom in—detailed images of the testis (top) and ovary (bottom) regions. FITC—FITC channel; DsRed—DsRed channel; BF—bright-field. Scale bars are 100 μm

method, leveraging the accumulated experience on transgenesis in other model organisms.

In this study, we demonstrate a reproducible transgenesis approach in *M. lignano* by microinjection and random integration of DNA constructs. Microinjection is the method of choice for creating transgenic animals in many species and allows delivery of the desired material into the egg, whether it is RNA, DNA, or protein[11]. Initially, we tried transposon- and

meganuclease-mediated approaches for integration of foreign DNA in the genome, but found in the course of the experiments that instead, random integration is a more efficient way for DNA incorporation in *M. lignano*. Random integration utilizes the molecular machinery of the host, integrating the provided DNA without the need for any additional components[39]. The method has its limitations, since the location and the number of integrated transgene copies cannot be controlled, and integration in a

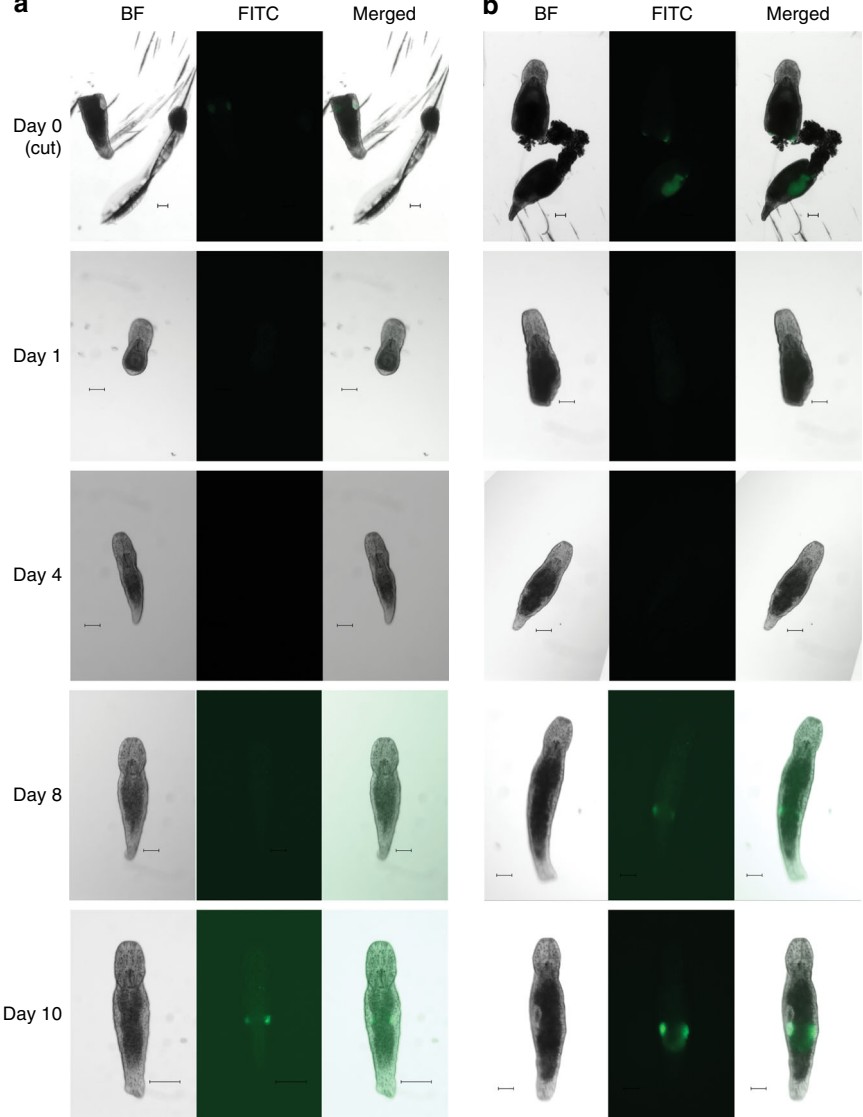

**Fig. 4** Transgene expression during regeneration. **a** Testes-specific transgenic line NL23. **b** Ovaries-specific transgenic line NL22. BF—bright-field, FITC—FITC channel. Day 0—animals immediately after amputation, both head and tail regions are shown. Only regenerating head regions are subsequently followed. Scale bars are 100 μm

functional site can cause unpredictable disturbances and variation in transgene expression[39]. Indeed, we observed differences in the expression levels between independent transgenic lines for the *EFA* transgene reporter (Fig. 5).Transgene silencing might occur in a copy-dependent manner, as is the case in the germline of *C. elegans*[40]. However, the fact that we readily obtained transgenic lines with germline-specific expression (Fig. 3c–e) indicates that germline transgene silencing is not a major issue in *M. lignano*.

The efficiency of integration and germline transmission varied between 1 and 8% of injected eggs in our experiments (Table 1), which is reasonable, given that a skilled person can inject up to 50 eggs in 1 h. Although injection of a circular plasmid carrying a transgene can result in integration and germline transmission with acceptable efficiency (e.g., line NL23, Table 1), we found that injection of vector-free[20] transgenes followed by ionizing irradiation of injected embryos with a dose of 2.5 Gy gave more consistent results (Table 1). Irradiation is routinely used in *C. elegans* for integration of extrachromosomal arrays, presumably by creating DNA breaks and inducing non-homologous recombination[19]. While irradiation can have deleterious consequences

by inducing mutations, in our experiments we have not observed any obvious phenotypic deviations in the treated animals and their progeny. Nevertheless, for the downstream genetic analysis involving transgenic lines, several rounds of backcrossing to non-irradiated stock might be required to remove any introduced mutations, which is easily possible given that these worms are outcrossing and have a short generation time[16,41]. Despite the mentioned limitations, random integration of foreign DNA appears to be a straightforward and productive approach for generating transgenic lines in *M. lignano* and can be used as a basis for further development of more controlled transgenesis methods in this animal, including transposon-based[42], integrase-based[43], homology-based[44], or CRISPR/Cas9-based[45] approaches.

The draft genome assembly of the *M. lignano* DV1 line, which is also used in this study, was recently published[9]. The genome appeared to be difficult to assemble and even the 130× coverage of PacBio data resulted in the assembly with N50 of only 64 Kb[9], while in other species N50 in the range of several megabases is usually achieved with such PacBio data coverages[46]. By adding

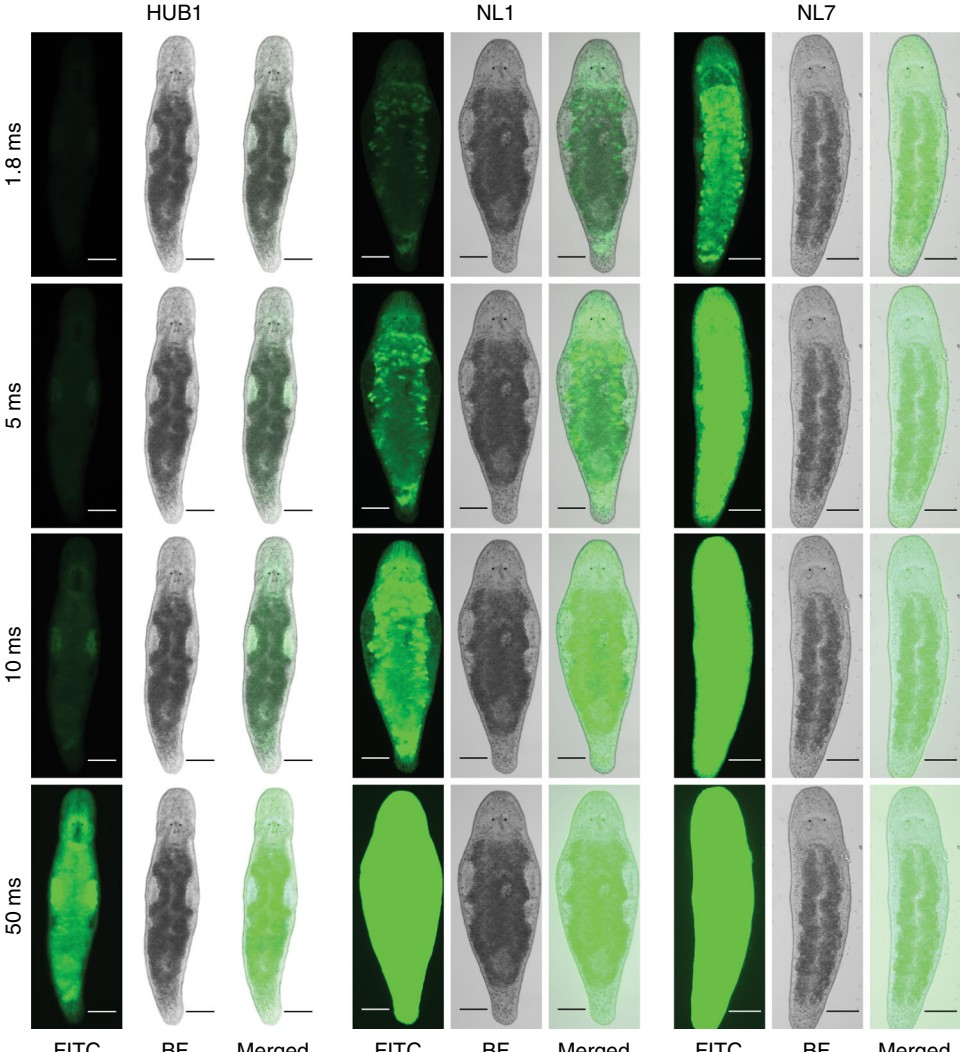

**Fig. 5** Variation of expression between different elongation factor 1 alpha transgenic lines. Fluorescence intensity is compared by taking images under the same exposure conditions at different exposure times (1.8 ms, 5 ms, 10 ms, and 50 ms). HUB1, NL1, NL7 – transgenic lines described in Table 1. FITC—FITC channel; BF—bright-field. Scale bars are 100 μm

Illumina and 454 data and using a different assembly algorithm, we have generated a substantially improved draft genome assembly, Mlig_3_7, with N50 scaffold size of 245.9 Kb (Table 2). The difficulties with the genome assembly stem from the unusually high fraction of simple repeats and transposable elements in the genome of *M. lignano*[9]. Furthermore, it was shown that *M. lignano* has a polymorphic karyotype and the DV1 line used for genome sequencing has additional large chromosomes (ref. [24] and Supplementary Fig. 3), which further complicates the assembly. The chromosome duplication also complicates genetic analysis and in particular gene knockout studies. To address these issues, we have established a different wild-type *M. lignano* line, NL10, from animals collected in the same geographical location as DV1 animals. The NL10 line appears to have no chromosomal duplications or they are present at a very low rate in the population, and its measured genome size is 500 Mb (Supplementary Fig. 3). While the majority of transgenic lines reported here are derived from the DV1 wild-type line, we observed similar transgenesis efficiency when using the NL10 line (Table 1, line NL24). Therefore, we suggest that NL10 line is a preferred line for future transgenesis applications in *M. lignano*.

To facilitate the selection of promoter regions for transgenic reporter constructs, we have generated Mlig_RNA_3_7

transcriptome assembly, which incorporates information from 5'- and 3'-specific RNA-seq libraries, as well as trans-splicing signals, to accurately define gene boundaries. We integrated genome assembly, annotation and expression data using the UCSC genome browser software (Supplementary Fig. 5, http://gb.macgenome.org). For genes tested in this study, the regions up to 2 kb upstream of the transcription start sites are sufficient to faithfully reflect tissue-specific expression patterns of these genes (Fig. 3), suggesting the preferential proximal location of gene regulatory elements, which will simplify analysis of gene regulation in *M. lignano* in the future.

In conclusion, we demonstrate that transgenic *M. lignano* animals can be generated with a reasonable success rate under a broad range of conditions, from circular and linear DNA fragments, with and without irradiation, as single and double reporters, and for multiple promoters, suggesting that the technique is robust. Similar to transgenesis in *C. elegans*, *Drosophila* and mouse, microinjection is the most critical part of the technique and requires skill that can be developed with practice. The generated genomic resources and the developed transgenesis approach provide a technological platform for harvesting the power of *M. lignano* as an experimental model organism for research on stem cells and regeneration.

 

## Methods

***M. lignano* lines and cultures**. The DV1 inbred *M. lignano* line used in this study was described previously[9,24,47]. The NL10 line was established from 5 animals collected near Lignano, Italy. Animals were cultured under laboratory conditions in plastic Petri dishes (Greiner), filled with nutrient enriched artificial sea water (Guillard's f/2 medium). Worms were fed ad libitum on the unicellular diatom *Nitzschia curvilineata* (Heterokontophyta, Bacillariophyceae) (SAG, Göttingen, Germany). Climate chamber conditions were set on 20 °C with constant aeration, a 14/10 h day/night cycle.

**Cloning of the elongation factor 1 alpha promoter**. The *M. lignano* EFA promoter sequence was obtained by inverse PCR. Genomic DNA was isolated using a standard phenol-chloroform protocol; fully digested by XhoI and subsequently self-ligated overnight (1 ng/µl). Diluted self-ligated gDNA was used for inverse PCR using the *EFA* specific primers Efa_IvPCR_rv3 5′-TCTCGAACTTCCACA-GAGCA-3′ and Efa_IvPCR_fw3 5′-CAAGAAGGAGGAGACCACCA-3′. Subsequently, nested PCR was performed using the second primer pair Efa_IvPCR_rv2 5′-AAGCTCCTGTGCCTCCTTCT-3′ and Efa_IvPCR_fw2 5′-AGGT-CAAGTCCGTCGAAATG-3′. The obtained fragment was cloned into p-GEM-T and sequenced. Later on, the obtained sequence was confirmed with the available genome data. Finally, the obtained promoter sequence was cloned into two different plasmids: the MINOS plasmid (using EcoRI/NcoI) and the I-SceI plasmid (using PacI/AscI).

**Codon optimization**. Highly expressed transcripts were identified from RNA-seq data[8] and codon weight matrices were calculated using the 100 most abundantly expressed non-redundant genes. *C. elegans* Codon Adapter code[48] was adapted for *M. lignano* (http://www.macgenome.org/codons) and used to design codon-optimized coding sequences (Supplementary Data 1). Gene fragments (IDT, USA) containing codon-optimized sequences, *EFA* 3′UTR and restriction cloning sites, were inserted into the pCS2+ vector to create optiMac plasmids used in the subsequent promoter cloning.

**Cloning of tissue-specific promoters**. Promoters were selected using Mlig_3_7, as well as several earlier *M. lignano* genome assemblies and MLRNA1509 transcriptome assembly[8]. RAMPAGE signal was used to identify the transcription start site and an upstream region of 1–2.5 kb was considered to contain the promoter sequence. An artificial ATG was introduced after the presumed transcription start site. This ATG was in-frame with the GFP of the target vector. The selected regions were cloned into optiMac vector using HindIII and BglII sites. Primers and cloned promoter sequences are provided in Supplementary Data 1.

**Preparation and collection of eggs**. Worms used for egg laying were kept in synchronized groups of roughly 500 per plate and transferred twice per week to prevent mixing with newly hatching offspring. The day before microinjections, around 1000 worms from 2 plates were kept together (to increase the number of eggs laid per plate) and transferred to plates with fresh f/2 medium and no food (to remove the leftover food from the digestive tracks of the animals as food debris can attach to the eggs and impair the microinjections by clogging needles and sticking to holders). On the day of the injections, worms were once again transferred to fresh f/2 without food to remove any debris and eggs laid overnight. Worms were kept in the dark for 3 h and then transferred to light. After 30 min in the light, eggs were collected using plastic pickers made from microloader tips (Eppendorf, Germany), placed on a glass slide in a drop of f/2 and aligned in a line for easier handling.

**Needle preparation**. Needles used in the microinjection procedure were freshly pulled using either borosilicate glass capillaries with filament (BF100-50-10, Sutter Instrument, USA) or aluminosilicate glass capillaries with filament (AF100-64-10, Sutter Instrument, USA) on a Sutter P-1000 micropipette puller (Sutter Instrument, USA) with the following settings: Heat = ramp-34, Pull = 50, Velocity = 70, Time = 200, Pressure = 460 for borosilicate glass and Heat = ramp, Pull = 60, Velocity = 60, Time = 250, Pressure = 500 for aluminosilicate glass. The tips of the needles were afterwards broken and sharpened using a MF-900 microforge (Narishige, Japan). Needles were loaded using either capillary motion or microloader tips (Eppendorf, Germany). Embryos were kept in position using glass holders pulled from borosilicate glass capillaries without a filament (B100-50-10, Sutter Instrument, USA) using P-1000 puller with the following settings: Heat = ramp + 18, Pull = 0, Velocity = 150, Time = 115, Pressure = 190. The holders were broken afterwards using a MF-900 microforge to create a tip of ~140 µm outer diameter and 50 µm inner diameter. Tips were heat-polished to create smooth edges and bent to a ~20° angle.

**Microinjections**. All microinjections were carried out on fresh one-cell stage *M. lignano* embryos. An AxioVert A1 inverted microscope (Carl Zeiss, Germany) equipped with a PatchMan NP2 for the holder and a TransferMan NK2 for the needle (Eppendorf, Germany) was used to perform all of the micromanipulations. A FemtoJet express (Eppendorf, Germany), with settings adjusted manually based on the amount of mucous and debris surrounding the embryos, was used as the pressure source for microinjections. A PiezoXpert (Eppendorf, Germany) was used to facilitate the penetration of the eggshell and the cell membrane of the embryo.

**Irradiation**. Irradiation was carried out using a IBL637 Caesium-137 source (CISbio International, France). Embryos were exposed to 2.5 Gy of γ-radiation within 1 h post injection.

**Establishing transgenic lines**. Positive hatchlings ($P_0$) were selected based on the presence of fluorescence and transferred into single wells of a 24-well plate. They were then crossed with single-wild-type worms that were raised in the same conditions. The pairs were transferred to fresh food every 2 weeks. Positive $F_1$ animals from the same $P_0$ cross were put together on fresh food and allowed to generate $F_2$ progeny. After the population of positive $F_2$ progeny grew to over 200 hatchlings, transgenic worms were singled out and moved to a 24-well plate. The selected worms were then individually back-crossed with wild-type worms to distinguish $F_2$ animals homozygous and heterozygous for the transgene. The transgenic $F_2$ worms that gave only positive progeny in the back-cross (at least 10 progeny observed) were assumed to be homozygous, singled out, moved to fresh food and allowed to lay eggs for another month to purge whatever remaining wild-type sperm from the back-cross. After the homozygous $F_2$ animals stopped producing new offspring, they were crossed to each other to establish a new transgenic line. The lines were named according to guidelines established at http://www.macgenome.org/nomenclature.html.

**Microscopy**. Images were taken using a Zeiss Axio Zoom V16 microscope with an HRm digital camera and Zeiss filter sets 38HE (FITC) and 43HE (DsRed), an Axio Scope A1 with a MRc5 digital camera or an Axio Imager M2 with an MRm digital camera.

**Southern blot analysis**. Southern blots were done using the DIG-System (Roche), according to the manufacturer's manual with the following parameters: vacuum transfer at 5 Hg onto positively charged nylon membrane for 2 h, UV cross-linking 0.14 J/cm², overnight hybridization at 68 °C.

**Identification of transgene integration sites**. The Universal GenomeWalker 2.0 Kit (Clontech Laboratories, USA) with restriction enzymes StuI and BamHI was used according to the manufacturer's protocol. Sanger sequencing of PCR products was performed by GATC Biotech (Germany).

**Whole mount in situ hybridization**. cDNA synthesis was carried out using the SuperScript III First-Strand Synthesis System (Life Technologies, USA), following the protocol supplied by the manufacturer. Two micrograms of total RNA were used as a template for both reactions: one with oligo(dT) primers and one with hexamer random primers. Amplification of selected DNA templates for ISH probes was performed by standard PCR with GoTaq Flexi DNA Polymerase (Promega, USA). Amplified fragments were cloned into pGEM-T vector system (Promega, USA) and validated by Sanger sequencing. Primers used for amplification are listed in Supplementary Data 1. Templates for riboprobes were amplified from sequenced plasmids using High Fidelity Pfu polymerase (Thermo Scientific, USA). pGEM-T backbone binding primers: forward (5′-CGGCCGCCATGGCCGCGGGA-3′) and reversed (5′-TGCAGGCGGCCGCACTAGTG-3′) and versions of the same primers with an upstream T7 promoter sequence (5′-GGATCCTAA-TACGACTCACTATAGG-3′. Based on the orientation of the insert in the vector either forward primer with T7 promoter and reverse without or vice versa, were used to amplify ISH probe templates. Digoxigenin (DIG) labeled RNA probe synthesis was performed using the DIG RNA labeling Mix (Roche, Switzerland) and T7 RNA polymerase (Promega, USA) following the manufacturer protocol. The concentration of all probes was assessed with the Qubit RNA BR assay (Invitrogen). Probes were then diluted in Hybridization Mix[49] (20 ng/µl), and stored at −80 °C. The final concentration of the probe and optimal hybridization temperature were optimized for every probe separately. Whole mount in situ hybridization was performed following a published protocol[49]. Pictures were taken using a standard light microscope with DIC optics and an AxioCam HRC (Zeiss, Germany) digital camera.

**Karyotyping**. DV1 and NL10 worms were cut above the testes and left to regenerate for 48 h to increase the amount of dividing cells[24]. Head fragments were collected and treated with 0.2% colchicine in f/2 (Sigma, C9754-100 mg) for 4 h at 20 °C to arrest cells in mitotic phase. Head fragments were then collected and treated with 0.2% KCl as hypotonic treatment for 1 h at room temperature. Fragments were then put on SuperfrostPlus slides (Fisher, 10149870) and macerated using glass pipettes while being in Fix 1 solution ($H_2O$: EtOH: glacial acetic acid 4:3:3). The cells were then fixed by treatment with Fix 2 solution (EtOH: glacial acetic acid 1:1) followed by Fix 3 solution (100% glacial acetic acid), before mounting by using Vectashield with Dapi (Vectorlabs, H-1200). At least three karyotypes were observed per worm and 20 worms were analyzed per line.

 

**Genome size measurements**. Genome size of the DV1 and NL10 lines was determined using flow cytometry approach[50]. In order eliminate the residual diatoms present in the gut, animals were starved for 24 h. For each sample 100 worms were collected in an Eppendorf tube. Excess f/2 was aspirated and worms were macerated in 200 μl 1× Accutase (Sigma, A6964-100ML) at room temperature for 30 min, followed by tissue homogenization through pipetting. 800 μl f/2 was added to the suspension and cells were pelleted by centrifugation at 4 °C, 1000 r.p. m., 5 min. The supernatant was aspirated and the cell pellet was resuspended in the nuclei isolation buffer (100 mM Tris-HCl pH 7.4, 154 mM NaCl, 1 mM $CaCl_2$, 0.5 mM $MgCl_2$, 0.2% BSA, 0.1% NP-40 in MilliQ water). The cell suspension was passed through a 35 μm pore size filter (Corning, 352235) and treated with RNase A and 10 mg/ml PI for 15 min prior to measurement. *Drosophila* S2 cells (gift from O. Sibon lab) and chicken erythrocyte nuclei (CEN, BioSure, 1006, genome size 2.5 pg) were included as references. The S2 cells were treated in the same way as *Macrostomum* cells. The CEN were resuspended in PI staining buffer (50 mg/ml PI, 0.6% NP-40 in calcium and magnesium free Dulbecco's PBS Life Technologies, 14190136). Fluorescence was measured on a BD FacsCanto II Cell Analyzer first separately for all samples and then samples were combined based on the amount of cells to obtain an even distribution of different species. The combined samples were re-measured and genome sizes calculated using CEN as a reference and S2 as positive controls (Supplementary Fig. 3).

**Preparation of genomic libraries**. One week prior to DNA isolation animals were kept on antibiotic-containing medium. Medium was changed every day with 50 μg/ml streptomycin or ampicillin added in alternating fashion. Worms were starved 24 h prior to extraction, and then rinsed in fresh medium. Genomic DNA was extracted using the USB PrepEase Genomic DNA Isolation kit (USB-Affymetrix, Cat. No. 78855) according to manufacturer's instructions. For the lysis step worms were kept in the supplied lysis buffer (with Proteinase K added) at 55 °C for 30–40 min and mixed by inverting the tube every 5 min. DNA was ethanol-precipitated once following the extraction and resuspended in TE buffer (for making 454 libraries Qiagen EB buffer was used instead). Concentration of DNA samples was measured with the Qubit dsDNA BR assay kit (Life Technologies, Cat. No. Q32850).

454 shotgun DNA libraries were made with the GS FLX Titanium General Library Preparation Kit (Roche, Cat. No. 05233747001), and for paired-end libraries the set of GS FLX Titanium Library Paired-End Adaptors (Roche, Cat. No. 05463343001) was used additionally. All the libraries were made following the manufacturer's protocol and sequenced on 454 FLX and Titanium systems.

Illumina paired-end genomic libraries were made with the TruSeq DNA PCR-free Library Preparation Kit (Ilumina, Cat. No. FC-121-3001) following the manufacturer's protocol. Long-range mate-pair libraries were prepared with the Nextera Mate Pair Sample Preparation Kit (Illumina, Cat. No. FC-132-1001) according to manufacturer's protocol. Libraries were sequenced on the Illumina HiSeq 2500 system.

**Genome assembly**. PacBio data (acc. SRX1063031) were assembled with Canu[25] v. 1.4 with default parameters, except the errorRate was set to 0.04. The resulting assembly was polished with Pilon[51] v. 1.20 using Illumina shotgun data mapped by Bowtie[52] v. 2.2.9 and RNA-seq data mapped by STAR[53] v. 2.5.2b. Next, scaffolding was performed by SSPACE[26] v. 3.0 using paired-end and mate-pair Illumina and 454 data. Mitochondrial genome of *M. lignano* was assembled separately from raw Illumina reads using the MITObim software[54] and the *Dugesia japonica* complete mitochondrial genome (acc. NC_016439.1) as a reference. The assembled mitochondrial genome differed from the recently published *M. lignano* mitochondrial genome[55] (acc. no. MF078637) in just 1 nucleotide in an intergenic spacer region. The genome assembly scaffolds containing mitochondrial sequences were filtered out and replaced with the separately assembled mitochondrial genome sequence. The final assembly was named Mlig_3_7. Genome assembly evaluation was performed with REAPR[27] and FRCbam[28] software using HUB1_300 paired-end library and DV1-6kb-1, HUB1-3_6 kb, HUB1-3_7 kb, ML_8KB_1 and ML_8KB_2 mate-pair libraries (Supplementary Table 3).

**Transcriptome assembly**. Previously published *M. lignano* RNA-seq data[8,31] (SRP082513, SRR2682326) and the de novo transcriptome assembly MLRNA150904 (ref. [8]) were used to generate an improved genome-guided transcriptome assembly. First, trans-splicing and polyA-tail sequences were trimmed from MLRNA150904 and the trimmed transcriptome was mapped to the Mlig_3_7 genome assembly by BLAT[56] v. 36 × 2 and hits were filtered using the pslCDna-Filter tool with the parameters "-ignoreNs -minId = 0.8 -globalNearBest = 0.01 -minCover = 0.95 −bestOverlap". Next, RNA-seq data were mapped to genome by STAR[53] v. 2.5.2b with parameters "--alignEndsType EndToEnd --twopassMode Basic --outFilterMultimapNmax 1000". The resulting bam files were provided to StringTie[29] v. 1.3.3 with the parameter "--rf", and the output was filtered to exclude lowly expressed antisense transcripts by comparing transcripts originating from the opposite strands of the same genomic coordinates and discarding those from the lower-expressing strand (at least fivefold read count difference). The filtered StringTie transcripts were merged with the MLRNA150904 transcriptome mappings using meta-assembler TACO[30] with parameters "--no-assemble-unstranded --gtf-expr-attr RPKM --filter-min-expr 0.01 --isoform-frac 0.75 --filter-min-length 100" and novel transcripts with RPKM <0.5 and not overlapping with MLRNA150904 mappings were discarded. The resulting assembled transcripts were termed 'Transcriptional Units' and the assembly named Mlig_R-NA_3_7_DV1.v1.TU. To reflect closely related transcripts in their names, sequences were clustered using cd-hit-est from the CD-HIT v. 4.6.1 package[57] with the parameters "-r 0 -c 0.95 -T 0 -M 0", and clustered transcripts were given the same prefix name. Close examination of the transcriptional units revealed that they often represented precursor mRNA for trans-splicing and contained several genes. Therefore, further processing of the transcriptional units to identified boundaries of the encoded genes was required. For this, we developed computational pipeline TBONE (Transcript Boundaries based ON experimental Evidence), which utilizes exclusively experimental data to determine precise 5′ and 3′ ends of trans-spliced mRNAs. Raw RNA-seq data were parsed to identify reads containing trans-splicing sequences, which were trimmed, and the trimmed reads were mapped to the genome assembly using STAR[53]. The resulting wiggle files were used to identify signal peaks corresponding to sites of trans-splicing. Similarly, for the identification of polyadenylation sites we used data generated previously[8] with CEL-seq library construction protocol and T-fill sequencing method. All reads originating from such an approach correspond to sequences immediately upstream of poly(A) tails and provide exact information on 3′UTR ends of mRNAs. The generated trans-splicing and poly(A) signals were overlapped with genomic coordinates of transcriptional units by TBONE, 'cutting' transcriptional units into processed mRNAs with exact gene boundaries, where such experimental evidence was available. Finally, coding potential of the resulting genes was estimated by TransDecoder[58], and transcripts containing ORFs but missing a poly(A) signal and followed by transcripts without predicted ORF but with poly(A) signal were merged if the distance between the transcripts was not >10 kb and the spanning region was repetitive. The resulting assembly was named Mlig_RNA_3_7_DV1.v1.genes and includes alternatively spliced and non-coding transcripts. To comply with strict requirements for submission of genome annotations to DDBJ/ENA/GenBank, the transcriptome was further filtered to remove alternative transcripts with identical CDS, and to exclude non-coding transcripts and transcripts overlapping repeat annotations. This final transcriptome assembly was named Mlig_RNA_3_7_DV1. v1.coregenes and used in annotation of the Mlig_3_7 genome assembly for submission to DDBJ/ENA/GenBank.

**Annotation of transposable elements and genomic duplications**. Two methods were applied to identify repetitive elements de novo both from the raw sequencing data and from the assembled scaffolds. Tedna software[59] v. 1.2.1 was used to assemble transposable element models directly from the repeated fraction of raw Illumina paired-end sequencing reads with the parameters "-k 31 -i 300 -m 200 -t 37 --big-graph = 1000". To mine repeat models directly from the genome assembly, RepeatModeler package (http://www.repeatmasker.org) was used with the default settings. Identified repeats from both libraries were automatically annotated using RepeatClassifier perl script from the RepeatModeler package against annotated repeats represented in the Repbase Update − RepeatMasker edition database[60] v. 20170127. Short (<200 bp) and unclassified elements were filtered out from both libraries. Additional specific de novo screening for full-length long terminal repeats (LTR) retrotransposons was performed using the LTRharvest tool[61] with settings "-seed 100 -minlenltr 100 -maxlenltr 3000 -motif tgca -mindistltr 1000 -maxdistltr 20000 -similar 85.0 -mintsd 5 -maxtsd 20 -motifmis 0 -overlaps all". Identified LTR retrotransposons were then classified using the RepeatClassifier perl script filtering unclassified elements. Generated repeat libraries were merged together with the RepeatMasker[60] library v. 20170127. The resulted joint library was mapped on the genome assembly with RepeatMasker. Tandem repeats were annotated and masked with Tandem Repeat Finder[62] with default settings. Finally, to estimate overall repeat fraction of the assembly, the Red de novo repeat annotation tool[63] with default settings was applied.

To identify duplicated non-repetitive fraction of the genome, repeat-masked genome assembly was aligned against itself using LAST software[64], and aligned non-self blocks longer than 500 nt and at least 95% identical were calculated.

**Data availability**. All raw data have been deposited in the NCBI Sequence Read Archive under accession codes SRX2866466 to SRX2866494. Annotated genome assembly has been deposited at DDBJ/ENA/GenBank under the accession NIVC00000000. The version described in this paper is version NIVC01000000. The genome and transcriptome assembly files are also available for download at http://gb.macgenome.org/downloads/Mlig_3_7.

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

## Acknowledgements

We thank H. Clevers for the support on the early stages of the project; E. Cuppen, E. de Bruin, P. van Zon and H. Lunstroo for the help with generating 454 data, and ERIBA sequencing facility for generating Illumina data. This work was supported by the

European Research Council (ERC Starting Grant "MacModel", grant no. 310765) to E.B., K.U. was supported by the project 0324-2016-0008 from the Russian State Budget. A.A. was supported by the Biotechnology and Biological Sciences Research Council (BBSRC, grant no. BB/K007564/1). P.L. was supported by the Austrian Science Fund (FWF, grant no. 25404). L.S. was supported by the Swiss National Science Foundation (SNFS, grant no. 3100A0-127503 and 31003A-143732). The work on annotation of transposable elements was supported by the Russian Foundation for Basic Research (RFBR, grant no. 15-04-08003) to E.B.

## Author contributions

E.B., P.L., and L.S. conceived the project. E.B. supervised the project and provided resources. J.Wudarski, K.d.M., T.D., D.S., P.W., M.Gre, and K.U. made constructs and performed transgenesis. J.Wudarski optimized transgenesis efficiency. L.G., F.B., M.Gre., M.Gru, and D.V. maintained *M. lignano* cultures. D.O. and L.G. established the NL10 line. D.S. and M.Gru generated genomic and RAMPAGE libraries. A.A., W.Q., L.S., E.B. contributed to sequencing genomic libraries. F.B. and S.M. performed genome size measurement and karyotyping. V.G. and E.B. performed genome and transcriptome assemblies and annotation. K.U. performed transposon annotation. J.Wudarski, D.S. and J.Wunderer performed in situ hybridizations. J.Wudarski and E.B. wrote the manuscript. All authors read the manuscript and provided the edits.

## Additional information

**Competing interests:** The authors declare no competing financial interests.

