## [Peer Review File · Nature Communications]

Reviewers' comments:

Reviewer #1 (Remarks to the Author):

The flatworm *Macrostomum lignano* has recently emerged as a promising model to study stem cell biology, germ cell biology, and regeneration. *M. lignano* researchers have a number of tools at their disposal including gene expression analysis by in situ hybridization and functional analysis by RNA interference. However, the molecular toolkit of this model organism is limited without the capacity for transgenesis. In this manuscript, Wudarski and colleagues report the successful development of transgenesis for *M. lignano*. While the authors have developed only a small number of stable transgenic lines, this work is an important resource for other flatworm biologists and has the potential to lead to the development of powerful experimental approaches including gene overexpression studies, live imaging, genetic lineage tracing, etc. The authors further facilitate the utility of this model organism by improving the current genome assembly and annotation and making it readily available to researchers on the UCSC genome browser.

The following suggestions should be addressed before publication:

1. Page 3 – In Fig. 1d, the authors show embryos expressing GFP after microinjection with GFP-encoding mRNA. The authors should show a no-GFP negative control to give readers a sense of how autofluorescent these embryos are (especially considering the fact that yolk is notoriously autofluorescent and can potentially obscure GFP signal). Negative controls should also be included in Figure 2 to help readers assess the level of autofluorescence in both the red and green channels.
2. Page 3 – The authors should refrain from using acronyms without defining them in the text. For example, what does HUB stand for (i.e. HUB1 transgenic line)? NL?
3. Page 4 – The authors state that they tested for the presence of a transiently expressed transgene (Supplementary Fig. 1f) by PCR – where is this data in the manuscript? Is it data not shown?
4. Figure 1g – The authors show a Southern blot depicting the presence of several copies of the *Efa::GFP* transgene but don't refer to this experiment in the text.

Reviewer #2 (Remarks to the Author):

Wudarski et al report on the development of transgenesis and associated genome assembly resources for *Macrostomum lignano* in this manuscript. *Macrostomum* is a flatworm related to planarians, and, like planarians is capable of regeneration in a process mediated by an adult proliferative stem cell population called neoblasts. Planarians have been the subjects of much molecular genetic investigation, but studies are limited by inability to generate transgenics. The results in this manuscript are clear - transgenesis is now established in *Macrostomum*. This is an important advance for using flatworms as models for studies of regeneration. Furthermore, the enhanced genome assembly will help facilitate utilizing these advances in the future by facilitating design of transgenes. The browser resources will also contribute to utilization of this system by a broad community. The following comments should be addressed:

Writing: The introduction could add some text to better connect this work/system to a broader biological audience, including stem cell and regenerative biologists.

Insertion into the genome: Were any southern (or other suitable approach) analyses done for any lines (such as in fig 2/3) generated with the final protocol, to determine whether the transgenes were indeed integrated into the genome and inserted at single or multiple locations? Showing integration into the genome would be essential. To enhance understanding of the method, was inverse PCR tried to test the site of integration? Similarly, are there any experiments that could determine whether the transgenes are single copy or multi-copy at a given insertion site?

Inheritance: In maintaining the lines, is transgene absence observed in some proportion of offspring? This is expected if animals are heterozygous for a single insertion. Similarly, the authors note the potential future importance of backcrossing lines because of the irradiation step and any associated mutational load. Can the transgenes successfully be propagated, with retained expression properties, by backcrossing (or by crossing into different genetic backgrounds)? This would also help in knowing whether transgenes are inserted at a single location.

Regeneration: Following amputation, do new tissues still display the same strength and tissue specificity of transgene expression? This could be assessed with the ELAV4, MYH6, or CABP7 transgenes for instance.

Some biological data would improve the significance/impact of the work. For instance some imaging (live/snapshots) of muscle regeneration or gonad regeneration with existing lines could be attempted to show utility of the methods.

Genome assembly: could more information be added about what this new assembly brings to the analysis of the *Macrostomum* genome. For instance with case study information about genes or promoters that were previously incomplete, absent, or unclear in prior assemblies?

Supplementary table 7: It is not clear the sequence used for the Efa 3'UTR used in making the transgenes.

Promoter selection and transgene features: were any common features noticed among the promoters that worked best for generating stable lines? Given the AT bias of the non-genic genome, were some promoters attempted that were not permissive for random insertion? Is information available on how size of transgene affected ability to integrate (from the existing attempts/data)?

Figure 1: A better (such as DIC) image of a *Macrostomum* zygote would be nice, but not essential (here or in the supplement).

Figure 1 b/c: The term "egg" is used in place of zygote or embryo at many points in the text. In some cases, that might be suitable if it is a fertilized egg with pronuclei, but at other points it might be more suitable to call it an embryo (for example, figure 1b legend "developing eggs")

Figure 1d: what is the % injected embryos with expression.

Reviewer #3 (Remarks to the Author):

Review: Wudarski et al., A platform for efficient transgenesis in *M. lignano*, a flatworm model organism for stem cell research

Summary of Key Results and Conclusions

The manuscript, "A platform for efficient transgenesis in *M. lignano*, a flatworm model organism for stem cell research" describes a set of newly developed techniques and resources for *M. lignano* that are anticipated to facilitate its use as a model organism for studies of stem cell biology. It is well-

written with a clear and strong voice. The techniques include methodology for the generation of transient and germline transgenics, while the resources include improved genome and transcriptome assemblies. Using this suite of tools, the authors generate a series of tissue-specific stable transgenic lines. Considered together, the work described in this manuscript provides a comprehensive set of resources for further studies in this model system.

Originality and significance

In developing a transgenesis method for *M. lignano*, the authors evaluated several well-known approaches that have been successfully used to create germline transgenics in a variety of other organisms. In doing so, they provide the novel finding that transgenesis within this species is also possible. The technique (injection, coupled with irradiation to promote integration) is only relatively successful in generating germline transgenics compared to other organisms (1-5% here, vs. 25+% in zebrafish). Nonetheless, it demonstrates that it can be achieved.

The updated *M. lignano* assembly reported in this manuscript is generated from a hybrid collection of sequencing reads: Illumina and 454 reads newly generated by the authors and PacBio reads already in the public databases (SRX1063031). The latter were previously assembled with the Celera assembler, but the authors now use the Canu assembler to generate a substantially more contiguous assembly, as based on contig N50. They integrate their new read data to provide a further, but more modest, increase contiguity with a scaffolding step (SSPACE) and presumably improve sequence accuracy (Pilon).

The authors also generate a new transcriptome for this organism, using their new assembly and a previously generated de novo transcriptome and RNAseq data set. As there is no comparison provided to the prior transcriptome, it is difficult to assess whether this represents a significant improvement over the existing resource. Nonetheless, independent evaluations of the transcriptome indicate that it is of high quality and thus likely to be of use.

Though the research described in this manuscript does not represent any new conceptual approaches with respect to transgenesis or genome/transcriptome assembly, the author's investigation and application of existing techniques to a relatively new model organism does much to further an understanding of its biology, promote its use and demonstrate its suitability for such a role. As the use of *M. lignano* for stem cell research is relatively recent, the impact these techniques and resources will have on the field as a whole is not yet clear. However, they do clearly offer new avenues for analysis and an understanding of flatworm regeneration.

Data and Methodology

Although the authors evaluated a variety of different transgenesis approaches in order to find the one giving the best results in *M. lignano*, there is little evidence that they attempted to optimize the technique. For example, does the concentration or length of injected DNA matter? How do more subtle changes in irradiation levels impact the rate of germline transmission? Without these sorts of data, it is difficult to assess how robust the technique is, and thus the likelihood it will succeed in the hands of others. The authors should address this point in the manuscript.

While contig N50 length is typically considered a proxy metric for assembly quality, it is important to recognize that increased contiguity may come at the expense of mis-joined sequences. Given that the new read data had only a modest effect on further increasing the assembly N50, it seems that it was the assembly method (Celera vs. Canu) that was responsible for the bulk of the improvement. Thus, the authors need to offer more (independent) assurance that the updated assembly actually represents an improvement over the prior version, and not badly joined sequences. Are there other studies the authors can cite in which the same datasets have been assembled with Celera and Canu

and various assembly metrics compared, demonstrating the superiority of Canu? Within their own study, the authors must also do more to demonstrate this. While the authors use the alignment of transcripts from the transcriptome assembly MLRNA150904 to evaluate the Mlig_3.5 assembly, there should be a complementary analysis of this same dataset on the ML2 assembly in order to demonstrate the impact of the increased N50. Extending this analysis to evaluate the number of fragmented and/or frameshifted gene models in each assembly would also provide a clearer view of the impact of the increased N50 (see PMID: 21731731). The use of an assembly comparison method, like FRCbam, would also offer a more well-rounded view of the differences between the assemblies, as would comparison to an independent dataset, such as BAC sequences or an optical map.

Appropriate use of stats

N/A

Appropriate use of references

Yes

Suggested improvements

Major issues

The manuscript is lacking the SRA identifiers for the newly generated Illumina and 454 reads, and also lacks the Assembly identifier for the Mlig_3.5 genome (e.g. GCA_\$). The BioProject identifier provided does not (yet?) provide pointers to these, so it cannot be confirmed that the data has been submitted to a public database. Likewise, what is the identifier for the transcriptome assembly?

The manuscript needs to provide some evidence or discussion regarding the robustness of the transgenesis technique (see comments above).

The manuscript needs to include an evaluation of the consequence of increasing the assembly N50 in the updated assembly (see comments above).

The manuscript needs to more clearly present the improvement of the new transcriptome over the prior one.

Minor corrections

Typo, p.5: "havea" should be "have a"

Typo, p.5: (Ref. 8) is not properly formatted

Supplementary Figure 1: Panel 1g is not referenced in the main text, and no explanation of "DV1" is provided in the legend.

Supplementary Figure 2d: M1-M4 are not well defined in the legend or main text. What are these?

Supplementary Figure 3: "using the UCSC genome browser" is not entirely correct language. The display was developed using UCSC genome browser software, but the Mlig_3.5 genome assembly is not available from the UCSC genome browser website.

Typo, Supplementary Table 4: "Shortes" should be "Shortest"

Conclusions

The authors present a well-written manuscript that outlines a suite of resources that are anticipated to further use of *M. lignano* as a model organism for stem cell research. The most novel findings come from the demonstration of transgenesis techniques, but the paper also offers an updated assembly

and transcriptome. However, the authors need to more convincingly demonstrate that the increased N50 of their assembly has functional consequences on annotation or that the assembly is considered improved by other metrics and likewise more clearly indicate how their new transcriptome is an improvement over the prior one. Without this, the significance and impact of the work is limited/unclear. As *M. lignano* is a relatively new model organism and not widely used, the impact of the manuscript hinges upon whether the findings will enable greater adoption of this system and/or promote new discovery by those who already work with it.

Response to Reviewers' comments

We thank the reviewers for thorough and constructive comments that helped to substantially improve the manuscript. Below we summarise the main changes to the revised manuscript and provide point-by-point response to the raised issues.

Main changes in the revised manuscript

1. A new version of the genome assembly is provided, Mlig_3_7, which is based on the latest version of the Canu assembler and has a further improved N50 size.
2. A comparison of the Mlig_3_7 and ML2 assemblies is performed with genome assembly evaluation tools, demonstrating the superiority of the Mlig_3_7 assembly in both continuity and base accuracy.
3. A new version of a genome-based transcriptome assembly is provided. The assembly more specifically incorporates experimental evidence on gene boundaries, in particular for trans-splicing transcripts.
4. An additional transgenic line with a double reporter construct is included, further strengthening our case for the robustness of the developed transgenesis approach.
5. The genome integration site for the transgenic NL21 line is established.
6. The re-establishment of transgene expression patterns after regeneration in testis- and ovary-specific transgenic lines is now documented.
7. Raw sequencing data are submitted to NCBI Short Read Archive under accession numbers SRX2866466- SRX2866494. Annotated genome assembly is submitted to NCBI GenBank under accession number NIVC00000000.

The major changes are highlighted in the revised manuscript.

Point-by-point response to the Reviewer's comments

Reviewer #1 (Remarks to the Author):

*The flatworm *Macrostomum lignano* has recently emerged as a promising model to study stem cell biology, germ cell biology, and regeneration. *M. lignano* researchers have a number of tools at their disposal including gene expression analysis by *in situ* hybridization and functional analysis by RNA interference. However, the molecular toolkit of this model organism is limited without the capacity for transgenesis. In this manuscript, Wudarski and colleagues report the successful development of transgenesis for *M. lignano*. While the authors have developed only a small number of stable transgenic lines, this work is an important resource for other flatworm biologists and has the potential to lead to the development of powerful experimental approaches including gene overexpression studies, live imaging, genetic lineage tracing, etc. The authors further facilitate the utility of this model organism by improving the current genome assembly and annotation and making it readily available to researchers on the UCSC genome browser.*

The following suggestions should be addressed before publication:

1. Page 3 – In Fig. 1d, the authors show embryos expressing GFP after microinjection with GFP-encoding mRNA. The authors should show a no-GFP negative control to give readers a sense of how autofluorescent these embryos are (especially considering the fact that yolk is notoriously

autofluorescent and can potentially obscure GFP signal). Negative controls should also be included in Figure 2 to help readers assess the level of autofluorescence in both the red and green channels.

We now provide a new Supplementary Figure 1, which shows very low autofluorescence levels detected only at very high exposure time in both green and red channels in embryos as well as in adult animals.

2. Page 3 – The authors should refrain from using acronyms without defining them in the text. For example, what does HUB stand for (i.e. HUB1 transgenic line)? NL?

HUB and NL are the names of transgenic lines. We adopted the approach of the *C. elegans* community, where all strains generated in a given lab have the same prefix (e.g. the initials of the PI), followed by a number. This work was initiated at Hubrecht institute, hence the arbitrary chosen HUB name. All other reported lines were generated after Berezikov lab moved to ERIBA, Groningen. We chose to name all the new and future lines with the prefix 'NL'. However, as agreed among *Macrostomum* researchers, the prefix can be chosen arbitrary and does not need to mean anything. To provide a list of line names currently in use in the *Macrostomum* community, we set up a dedicated web page at <http://www.macgenome.org/nomenclature.html>. We added the following text in Materials and Methods: “The lines were named according to guidelines established at <http://www.macgenome.org/nomenclature.html>”.

3. Page 4 – The authors state that they tested for the presence of a transiently expressed transgene (Supplementary Fig. 1f) by PCR – where is this data in the manuscript? Is it data not shown?

Unfortunately, we do not have these results properly documented (which were essentially images of empty gels). Since references to ‘data not shown’ are not allowed at Nature Communications, we removed the sentence in question altogether. We would like to emphasise that this does not in any way influence the flow and the conclusions of the manuscript.

4. Figure 1g – The authors show a Southern blot depicting the presence of several copies of the *Efa::GFP* transgene but don't refer to this experiment in the text.

We added the following sentence in the main text: “Furthermore, Southern blot analysis revealed that HUB1 contains multiple transgene copies (Supplementary Fig. 2g).”

Reviewer #2 (Remarks to the Author):

Wudarski et al report on the development of transgenesis and associated genome assembly resources for Macrostomum lignano in this manuscript. Macrostomum is a flatworm related to planarians, and, like planarians is capable of regeneration in a process mediated by an adult proliferative stem cell population called neoblasts. Planarians have been the subjects of much molecular genetic investigation, but studies are limited by inability to generate transgenics. The results in this manuscript are clear - transgenesis is now established in Macrostomum. This is an important advance for using flatworms as models for studies of regeneration. Furthermore, the enhanced genome assembly will help facilitate utilizing these advances in the future by facilitating design of transgenes. The browser resources will also contribute to utilization of this system by a broad community.

The following comments should be addressed:

Writing: The introduction could add some text to better connect this work/system to a broader biological audience, including stem cell and regenerative biologists.

We added some new text and two references to reflect in more detail the current state of stem cell research in flatworms.

Insertion into the genome: Were any southern (or other suitable approach) analyses done for any lines (such as in fig 2/3) generated with the final protocol, to determine whether the transgenes were indeed integrated into the genome and inserted at single or multiple locations? Showing integration into the genome would be essential. To enhance understanding of the method, was inverse PCR tried to test the site of integration? Similarly, are there any experiments that could determine whether it transgenes are single copy or multi-copy at a given insertion site?

We now added the new Supplementary Figure 6 dedicated to establishing the integration site of the NL21 (ELAV4::GFP) transgenic line. We show that the inverse PCR approach is confounded by the presence of tandem transgene copies, and as a result the junction regions of the transgene are amplified, rather than junctions with genomic sequences. We next used Genome Walker approach, which involves ligation of adapters and can overcome the issue of tandem repeats. With this approach we successfully identified one side of transgene integration in the genome in NL21 line. We hence directly demonstrate that the transgene is integrated in the genome, but we did not pursue this approach for other transgenic lines: the Genome Walker approach requires very significant amounts of genomic DNA and trials with multiple restriction enzymes – a very substantial time and resources drain that we think in this case is not justified by the goal of establishing where specifically a potentially complex transgene array is integrated in a particular transgenic line.

Inheritance: In maintaining the lines, is transgene absence observed in some proportion of offspring? This is expected if animal are heterozygous for a single insertion. Similarly, the authors note the potential future importance of backcrossing lines because of the irradiation step and any associated mutational load. Can the transgenes successfully be propagated, with retained expression properties, by backcrossing (or by crossing into different genetic backgrounds)? This would also help in knowing whether transgenes are inserted at a single location.

As described in the Materials and Methods section “Establishing transgenic lines”, the very first step is backcrossing, i.e. crossing the positive transgenic animals with wild-type animals of the same genetic background. The progeny of these crosses are heterozygous for the transgene, and subsequent work is done with these progeny, as was done in two recently published studies based on back-crossing of the HUB1 line onto a range of different genetic backgrounds (Evolution 70:314-328; Evolution 71:1232–1245). Hence, transgenes definitely can propagate through backcrosses, and we think we provide sufficient evidence for this.

Concerning the long-term transgene propagation, in the section describing HUB1 line, we write “From these experiments we established the HUB1 transgenic line with ubiquitous GFP expression (Supplementary Fig. 2e), for which stable transgene transmission has been observed for over 50 generations^{16,17}”. We should note that the analysis of transgene inheritance patterns is complicated by the presence of karyotype polymorphism in the DV1 line, which has duplicated larger chromosomes, and the duplicated chromosome can be spontaneously lost. We do observe occasional transgene loss in the DV1-background lines but it is difficult to distinguish without additional significant efforts whether the loss is transgene-specific or due to the loss of the whole transgene-containing chromosome.

Regeneration: Following amputation, do new tissue still display the same strength and tissue specificity of transgene expression? This could be assessed with the ELAV4, MYH6, or CABP7 transgenes for instance. Some biological data would improve the significance/impact of the work. For

instance some imaging (live/snapshots) of muscle regeneration or gonad regeneration with existing lines could be attempted to show utility of the methods.

We now provide the new Supplementary Figure 7, where we monitored transgene expression in testis- and ovary-specific transgenic lines during regeneration. In both cases transgene expression is fully and specifically restored as expected from a regular genomic locus.

*Genome assembly: could more information be added about what this new assembly brings to the analysis of the *Macrostomum* genome. For instance with case study information about genes or promoters that were previously incomplete, absent, or unclear in prior assemblies?*

In line with this remark and also with the suggestion of Reviewer #3, we performed comparisons between our assembly and the previously published ML2 genome assembly using the genome assembly evaluation tools REAPR and FRCbam, as well as mapping of the *de novo* transcriptome. The results are summarized in the new Supplementary Figure 4, which demonstrates that the Mlig_3_7 assembly is superior to ML2 in both continuity and base accuracy.

Supplementary table 7: It is not clear the sequence used for the Efa 3'UTR used in making the transgenes.

The EFA 3'UTR sequence is added to the table.

Promoter selection and transgene features: were any common features noticed among the promoters that worked best for generating stable lines? Given the AT bias of the non-genic genome, were some promoters attempted that were not permissive for random insertion? Is information available on how size of transgene affected ability to integrate (from the existing attempts/data)?

The existing very limited data do not allow drawing general conclusions about common features of successful promoters. As for the transgene size, we now describe a new transgenic line based on a double-reporter construct, which is 2 times larger than the previous constructs but demonstrated similar efficiency (Fig. 3e). Also, we now provide evidence that transgenes integrate as tandem duplications, at least in the two studied transgenic lines (Supplementary Figure 6). Therefore, the size of the transgene is most likely not a major limitation for genomic integration.

*Figure 1: A better (such as DIC) image of a *Macrostomum* zygote would be nice, but not essential (here or in the supplement).*

We have added a DIC image of a zygote in Fig. 1c.

Figure 1 b/c: The term "egg" is used in place of zygote or embryo at many points in the text. In some cases, that might be suitable if it is a fertilized egg with pronuclei, but at other points it might be more suitable to call it an embryo (for example, figure 1b legend "developing eggs")

Corrected.

Figure 1d: what is the % injected embryos with expression.

We modified the following sentence to include this information: "Next, we injected in vitro synthesized mRNA encoding green fluorescent protein (GFP) and observed its expression in all successfully injected embryos (n > 100) within 3 hours after injection (Fig. 1e), with little to no autofluorescence detected in either embryos or adult animals (Supplementary Fig. 1)".

Reviewer #3 (Remarks to the Author):

Review: Wudarski et al., *A platform for efficient transgenesis in M. lignano, a flatworm model organism for stem cell research*

Summary of Key Results and Conclusions

The manuscript, "A platform for efficient transgenesis in *M. lignano*, a flatworm model organism for stem cell research" describes a set of newly developed techniques and resources for *M. lignano* that are anticipated to facilitate its use as a model organism for studies of stem cell biology. It is well-written with a clear and strong voice. The techniques include methodology for the generation of transient and germline transgenics, while the resources include improved genome and transcriptome assemblies. Using this suite of tools, the authors generate a series of tissue-specific stable transgenic lines. Considered together, the work described in this manuscript provides a comprehensive set of resources for further studies in this model system.

Originality and significance

In developing a transgenesis method for *M. lignano*, the authors evaluated several well-known approaches that have been successfully used to create germline transgenics in a variety of other organisms. In doing so, they provide the novel finding that transgenesis within this species is also possible. The technique (injection, coupled with irradiation to promote integration) is only relatively successful in generating germline transgenics compared to other organisms (1-5% here, vs. 25+% in zebrafish). Nonetheless, it demonstrates that it can be achieved.

The updated *M. lignano* assembly reported in this manuscript is generated from a hybrid collection of sequencing reads: Illumina and 454 reads newly generated by the authors and PacBio reads already in the public databases (SRX1063031). The latter were previously assembled with the Celera assembler, but the authors now use the Canu assembler to generate a substantially more contiguous assembly, as based on contig N50. They integrate their new read data to provide a further, but more modest, increase contiguity with a scaffolding step (SSPACE) and presumably improve sequence accuracy (Pilon).

The authors also generate a new transcriptome for this organism, using their new assembly and a previously generated *de novo* transcriptome and RNAseq data set. As there is no comparison provided to the prior transcriptome, it is difficult to assess whether this represents a significant improvement over the existing resource. Nonetheless, independent evaluations of the transcriptome indicate that it is of high quality and thus likely to be of use.

Though the research described in this manuscript does not represent any new conceptual approaches with respect to transgenesis or genome/transcriptome assembly, the author's investigation and application of existing techniques to a relatively new model organism does much to further an understanding of its biology, promote its use and demonstrate its suitability for such a role. As the use of *M. lignano* for stem cell research is relatively recent, the impact these techniques and resources will have on the field as a whole is not yet clear. However, they do clearly offer new avenues for analysis and an understanding of flatworm regeneration.

Data and Methodology

Although the authors evaluated a variety of different transgenesis approaches in order to find the one giving the best results in *M. lignano*, there is little evidence that they attempted to optimize the technique. For example, does the concentration or length of injected DNA matter? How do more subtle changes in irradiation levels impact the rate of germline transmission? Without these sorts of data, it is difficult to assess how robust the technique is, and thus the likelihood it will succeed in the hands of others. The authors should address this point in the manuscript.

While we did play with DNA concentrations and other parameters in the beginning of the project, we quickly realized that the central and most critical part of the technique is microinjection. This is a fully manual procedure and its outcome greatly depends on the skill and experience of the person performing the microinjection. This skill can be developed through exercise, but the procedure inherently cannot be fully standardized. For example, the amount of material delivered in a given microinjection shot will be always variable to some degree, as will be the depth of the needle penetration into the embryo. This variability of the microinjection procedure to a large extent voids more subtle optimizations, such as DNA concentrations. Importantly, we demonstrate throughout the

manuscript that transgenic lines in *M. lignano* can be generated under a broad range of conditions: with circular and linear constructs, with and without irradiation, with single and double reporters, and with different promoters. We therefore have every reason to think that the technique itself is rather robust, and its success in the hands of others would fully depend on their microinjection skills – same as for transgenesis in *C. elegans*, *Drosophila* or mouse. To reflect this, we added the following text in the Discussion: “In conclusion, we demonstrate that transgenic *M. lignano* animals can be generated with a reasonable success rate under a broad range of conditions, from circular and linear DNA fragments, with and without irradiation, as single and double reporters, and for multiple promoters, suggesting that the technique is robust. Similar to transgenesis in *C. elegans*, *Drosophila* and mouse, microinjection is the most critical part of the technique and requires skill that can be developed with practice.”

While contig N50 length is typically considered a proxy metric for assembly quality, it is important to recognize that increased contiguity may come at the expense of mis-joined sequences. Given that the new read data had only a modest effect on further increasing the assembly N50, it seems that it was the assembly method (Celera vs. Canu) that was responsible for the bulk of the improvement. Thus, the authors need to offer more (independent) assurance that the updated assembly actually represents an improvement over the prior version, and not badly joined sequences. Are there other studies the authors can cite in which the same datasets have been assembled with Celera and Canu and various assembly metrics compared, demonstrating the superiority of Canu? Within their own study, the authors must also do more to demonstrate this. While the authors use the alignment of transcripts from the transcriptome assembly MLRNA150904 to evaluate the Mlig_3.5 assembly, there should be a complementary analysis of this same dataset on the ML2 assembly in order to demonstrate the impact of the increased N50. Extending this analysis to evaluate the number of fragmented and/or frameshifted gene models in each assembly would also provide a clearer view of the impact of the increased N50 (see PMID: 21731731). The use of an assembly comparison method, like FRCbam, would also offer a more well-rounded view of the differences between the assemblies, as would comparison to an independent dataset, such as BAC sequences or an optical map.

We performed comparisons between our assembly and the previously published ML2 genome assembly using genome assembly evaluation tools REAPR and FRCbam, as well as mapping of the *de novo* transcriptome. The results are summarized in the new Supplementary Figure 4, which demonstrates that the Mlig_3_7 assembly is superior to ML2 in both continuity and base accuracy. Also, the paper describing the Canu assembler was recently published, demonstrating its superiority to other assemblers on multiple datasets.

The parts concerning the transcriptome assembly are now substantially revisited, as we took a novel approach to defining gene boundaries, particularly in case of trans-spliced transcripts. We think the improvements are now clear from the text.

Appropriate use of stats

N/A

Appropriate use of references

Yes

Suggested improvements

Major issues

The manuscript is lacking the SRA identifiers for the newly generated Illumina and 454 reads, and also lacks the Assembly identifier for the Mlig_3.5 genome (e.g. GCA_\$). The BioProject identifier provided does not (yet?) provide pointers to these, so it cannot be confirmed that the data has been submitted to a public database. Likewise, what is the identifier for the transcriptome assembly?

All data were submitted to NCBI with the immediate release date. The following section was added to the text: "All raw data were submitted to NCBI Sequence Read Archive under accession numbers SRX2866466- SRX2866494. Annotated genome assembly was submitted to GenBank under accession number NIVC00000000."

As of June 28, 2017, the raw data are publicly available. The annotated genome was submitted for the immediate release but still undergoes review by the NCBI staff. It will be released as soon as it passes the review. Please see the email from NCBI at the end of this letter.

Furthermore, there is no separate transcriptome submission/accession, as it forms the part of the annotated genome submission (the other part of the annotation is transposable elements). Essentially, we did not submit a bare genome assembly like it is common for many current WGS project but instead provided its proper annotation as well.

The manuscript needs to provide some evidence or discussion regarding the robustness of the transgenesis technique (see comments above).

Please see the reply in the respective comments section above.

The manuscript needs to include an evaluation of the consequence of increasing the assembly N50 in the updated assembly (see comments above).

Please see the reply in the respective comments section above.

The manuscript needs to more clearly present the improvement of the new transcriptome over the prior one.

Please see the reply in the respective comments section above.

Minor corrections

Typo, p.5: "havea" should be "have a"

Corrected

Typo, p.5: (Ref. 8) is not properly formatted

Corrected

Supplementary Figure 1: Panel 1g is not referenced in the main text, and no explanation of "DV1" is provided in the legend.

We added the following sentence: "Furthermore, Southern blot analysis revealed that HUB1 contains multiple transgene copies (Supplementary Fig. 2g)." Note that the numbering of the supplementary figures has changed, as we added additional figures.

Supplementary Figure 2d: M1-M4 are not well defined in the legend or main text. What are these?

Added in the legend: "M1-M5, gates used to calculate peak intensities of different genomes in the samples".

Supplementary Figure 3: "using the UCSC genome browser" is not entirely correct language. The display was developed using UCSC genome browser software, but the Mlig_3.5 genome assembly is not available from the UCSC genome browser website.

Corrected

Typo, Supplementary Table 4: “Shortes” should be “Shortest”

Corrected

Conclusions

*The authors present a well-written manuscript that outlines a suite of resources that are anticipated to further use of *M. lignano* as a model organism for stem cell research. The most novel findings come from the demonstration of transgenesis techniques, but the paper also offers an updated assembly and transcriptome. However, the authors need to more convincingly demonstrate that the increased N50 of their assembly has functional consequences on annotation or that the assembly is considered improved by other metrics and likewise more clearly indicate how their new transcriptome is an improvement over the prior one. Without this, the significance and impact of the work is limited/unclear. As *M. lignano* is a relatively new model organism and not widely used, the impact of the manuscript hinges upon whether the findings will enable greater adoption of this system and/or promote new discovery by those who already work with it.*

We believe that in the revised version of the manuscript we convincingly demonstrate the improvements to both genome and transcriptome assemblies. The most significant part of the work is however not the genome/transcriptome but the demonstration of transgenesis in *M. lignano*. We think that this work will be a turning point for *Macrostomum* as a model organism, as it will convince many new groups to adopt it for their research questions. We envision that *Macrostomum* will become “*C. elegans* of stem cell research”, and this work demonstrating transgenesis and opening up genomic resources for *M. lignano* will have significant impact for this development.

REVIEWERS' COMMENTS:

Reviewer #1 (Remarks to the Author):

The authors satisfactorily addressed all my comments and I think that it is acceptable for publication.

Reviewer #2 (Remarks to the Author):

The authors have addressed prior comments well.

Prior point 1: The suggestion was more to add a sentence or two that would mention types of stem cell biology questions that could be explored with transgenesis (for example, live imaging stem cell behavior, study of stem cell fate choice, etc.). This is a stylistic comment, up to the authors.

Typo:

Line 73 "neoblasts population" should read "neoblast population"

Reviewer #3 (Remarks to the Author):

The authors have added substantial new data in response to reviewer comments. These new data do not substantially change the overall conclusions of the manuscript, but instead offer further support for their arguments and contribute to more robust genomic/transcriptomic resources.

The authors have sufficiently addressed my major concerns with the manuscript. I applaud their more in-depth analysis of the quality of their improved genome assembly to demonstrate its superiority over the prior version. The manuscript now also provides the appropriate database identifiers for the datasets used and sequences submitted.

The transcriptome described in the revised manuscript was generated using a novel approach, using experimental data to more accurately define gene boundaries. The comprehensive analysis of this new transcriptome demonstrates its high quality and comparison to the transcriptome from the prior submission suggests the accommodations made to address trans-splicing have improved the dataset. Providing the authors can address the relatively minor points noted below, I support the publication of this manuscript.

Supplemental Fig 5: The font in this figure is too small. I suggest it be provided in landscape, rather than portrait, orientation.

P7, line 252: Extra space in "Table 1"

P11, line 427: "where" should be "were"

P18, line 671: Reference #8 is improperly cited. This should be eLife 2016; 5: e20607

Response to Reviewers' comments

We thank the reviewers for the support and comments on the revised manuscript. No further major issues were raised, and we address the remaining comments as summarised below.

There is one change we introduce in the second manuscript revision. We realized that in our transcriptome assembly we were missing some important genes, such as TERT, due to the step of filtering out very lowly expressed transcripts early on in the assembly pipeline. We now made an adjustment to retain such low-level transcripts if they contain a predicted open reading frame of at least 100 amino-acids. As a result, we now provide a revised transcriptome assembly that contains more transcripts. We also now used additional RNA-seq data that became publicly available during the revision of the manuscript. The new transcriptome assembly version has a lower TransRate score compared to the previous version due to inclusion of the low-expression transcripts. Nevertheless, we believe that this is an improved and more representative transcriptome assembly.

Point-by-point response to the Reviewer's comments

Reviewer #1 (Remarks to the Author):

The authors satisfactorily addressed all my comments and I think that it is acceptable for publication.

Reviewer #2 (Remarks to the Author):

The authors have addressed prior comments well.

Prior point 1: The suggestion was more to add a sentence or two that would mention types of stem cell biology questions that could be explored with transgenesis (for example, live imaging stem cell behavior, study of stem cell fate choice, etc.). This is a stylistic comment, up to the authors.

We do discuss applications of transgenesis mentioned by the reviewer at the end of the Introduction.

Typo:

Line 73 "neoblasts population" should read "neoblast population"

Corrected

Reviewer #3 (Remarks to the Author):

The authors have added substantial new data in response to reviewer comments. These new data do not substantially change the overall conclusions of the manuscript, but instead offer further support for their arguments and contribute to more robust genomic/transcriptomic resources.

The authors have sufficiently addressed my major concerns with the manuscript. I applaud their more in-depth analysis of the quality of their improved genome assembly to demonstrate its superiority over the prior version.

The manuscript now also provides the appropriate database identifiers for the datasets used and sequences submitted.

The transcriptome described in the revised manuscript was generated using a novel approach, using experimental data to more accurately define gene boundaries. The comprehensive analysis of this new transcriptome demonstrates its high quality and comparison to the transcriptome from the prior submission suggests the accommodations made to address trans-splicing have improved the dataset.

Providing the authors can address the relatively minor points noted below, I support the publication of this manuscript.

Supplemental Fig 5: The font in this figure is too small. I suggest it be provided in landscape, rather than portrait, orientation.

We made a new Supplemental Fig.5 with increased font size. All details are now readable even in the preferred portrait orientation.

P7, line 252: Extra space in "Table 1"

Corrected.

P11, line 427: "where" should be "were"

Corrected.

P18, line 671: Reference #8 is improperly cited. This should be eLife 2016; 5: e20607

Corrected.